# New Insights into Tetraplosphaeriaceae Based on Taxonomic Investigations of Bambusicolous Fungi and Freshwater Fungi

**DOI:** 10.3390/jof10050319

**Published:** 2024-04-27

**Authors:** Hai-Jun Zhao, Mingkwan Doilom, Ausana Mapook, Gennuo Wang, Kevin D. Hyde, Wei Dong

**Affiliations:** 1Innovative Institute for Plant Health/Key Laboratory of Green Prevention and Control on Fruits and Vegetables in South China, Ministry of Agriculture and Rural Affairs, Zhongkai University of Agriculture and Engineering, Guangzhou 510225, China; zhaohaijun2020@163.com (H.-J.Z.); kdhyde3@gmail.com (K.D.H.); 2Center of Excellence in Fungal Research, Mae Fah Luang University, Chiang Rai 57100, Thailand; ausana.map@mfu.ac.th; 3School of Science, Mae Fah Luang University, Chiang Rai 57100, Thailand; 4Johann Friedrich Blumenbach Institute of Zoology and Anthropology, University of Göttingen, 37073 Göttingen, Germany; gennuo.wang@uni-goettingen.de; 5Department of Botany and Microbiology, College of Science, King Saud University, Riyadh 11362, Saudi Arabia; 6Yunnan Key Laboratory of Fungal Diversity and Green Development, Key Laboratory for Plant Diversity and Biogeography of East Asia, Kunming Institute of Botany, Chinese Academy of Sciences, Kunming 650201, China

**Keywords:** bamboo, compilation, host and habitat specificity, new species, submerged wood, taxonomy

## Abstract

Species within Tetraplosphaeriaceae have been frequently documented in recent years with the extensive investigations of microfungi along a latitudinal gradient from north to south in the Asian/Australian region. Both bamboo substrates and freshwater habitats serve as extensive reservoirs, hosting a rich diversity of fungi that exhibit broad geographical distributions. The most common fungi in these two environments are generally distributed in distinct families. However, our statistics have revealed an intriguingly distinct preference of Tetraplosphaeriaceae species for inhabiting both bamboo substrates and freshwater habitats. The genera *Pseudotetraploa* (100%) and *Triplosphaeria* (100%) exhibit a strong preference, followed by *Shrungabeeja* (71%) and *Quadricrura* (67%). Our taxonomic and phylogenetic study of microfungi in southern China have identified four additional novel species, viz., *Aquatisphaeria bambusae* sp. nov., *Pseudotetraploa phyllostachydis* sp. nov., *Pseudotetraploa yangjiangensis* sp. nov., and *Tetraploa submersa* sp. nov. from bamboo substrates and freshwater habitats. In addition, *Aquatisphaeria thailandica* has previously been documented from freshwater habitats in Thailand; however, we have once again isolated this species from decaying bamboo substrates in Guangdong, China. The new findings substantiate our hypothesis that the preference of Tetraplosphaeriaceae species for colonizing bamboo substrates and freshwater habitats will be more evident through more extensive investigations conducted in such environments.

## 1. Introduction

Tetraplosphaeriaceae was introduced by Tanaka et al. [1] to accommodate five tetraploa-like genera, viz., *Polyplosphaeri*, *Pseudotetraploa*, *Quadricrura*, *Tetraplosphaeria* (type), and *Triplosphaeria*. In a reassessment conducted by Hyde et al. [2], *Tetraplosphaeria* was treated as a synonym of *Tetraploa*, which had previously been applied to the asexual state. Therefore, *Tetraploa* now represents the type of the family. Subsequently, Ariyawansa et al. [3] and Dong et al. [4] incorporated *Shrungabeeja* and *Ernakulamia* into their phylogenetic analyses. Based on morphological similarities and molecular characteristics, the inclusion of these two genera in the family is widely accepted [5,6,7,8]. According to the “Outline of Fungi and fungus-like taxa—2021”, a total of nine genera have been listed within the family, viz., *Aquatisphaeria*, *Byssolophis*, *Ernakulamia*, *Polyplosphaeria*, *Pseudotetraploa*, *Quadricrura*, *Shrungabeej*, *Tetraploa*, and *Triplosphaeria* [9]. Recently, Zhang et al. [10] introduced one additional new genus, *Pseudopolyplosphaeria*, which was collected from karst landscapes of Guizhou Province on dead bamboo culms. Hyde et al. [2] and Dong et al. [4] have provided a taxonomic key for several genera within Tetraplosphaeriaceae. Tetraplosphaeriaceae is characterized by massarina-like sexual morphs with almost hyaline, 1(–3)-septate ascospores and/or tetraploa-like asexual morphs with several setose appendages [1]. The majority of species within the family have been recorded as saprobic, primarily associated with bamboo [1,11,12]. Additionally, some species have also been found to inhabit aquatic environments [4,5,7,8,13].

The type genus *Tetraploa* was introduced by Berkeley and Broome [14] with *T. aristata* as the type species. The sexual morph is characterized by immersed to erumpent, globose to subglobose, glabrous ascomata, cylindrical to clavate, short pedicellate asci, and narrowly fusiform, septate ascospores with a narrow mucilaginous appendage-like sheath [1]. The asexual morph is characterized by cylindrical, brown, 4-columned conidia with 4-setose, divergent, short apical appendages [1]. To date, 38 species epithets have been listed in Index Fungorum (http://www.indexfungorum.org/names/names.asp, accessed on 24 February 2024) [15]. Most species, including the type species *T. aristata*, have been subjected to sequencing and subsequently incorporated into phylogenetic analyses [4,5,6,7]. The genus *Pseudotetraploa*, typified by *P. curviappendiculata*, has consistently shown close affinities with *Tetraploa* in previous studies [16,17]. They can be distinguished by their conidial shape (short cylindrical vs. obpyriform), septation (eusepta vs. pseudosepta), and appendages (long and straight vs. short and curved) [1]. Based on these morphological differences, three species of *Tetraploa*, viz., *T. curviappendiculata*, *T. javanica*, and *T. longissima*, were transferred to *Pseudotetraploa*, which was supported by molecular evidence [1]. Among all six species, with the exception of *P. yunnanensis* which was collected from freshwater habitats [7], the remaining species were obtained from terrestrial habitats [6,17,18]. All species form a monophyletic clade in the family Tetraplosphaeriaceae. Another tetraploa-like genus, *Aquatisphaeria*, was established for a freshwater species *A. thailandica*, which was isolated from submerged wood in Thailand [5]. The sexual morph of *Aquatisphaeria* remains undetermined, while the asexual morph is characterized by macronematous, cylindrical conidiophores, subglobose or turbinate, dictyoseptate, pale olivaceous to dark brown conidia with cylindrical appendages [5].

We are studying the freshwater microfungi along a latitudinal gradient from north to south in the Asian/Australian region [19]. As bamboo grows alongside rivers, the fungi found on both are very much intertwined [20]. In the present study, we investigate the tetraploa-like taxa from decaying bamboo and submerged wood in Southern China, and revise the taxon diversity within Tetraplosphaeriaceae. A total of five specimens were subjected to morphological and phylogenetic analyses, and the preliminary results indicated that they belong in *Aquatisphaeria*, *Pseudotetraploa*, and *Tetraploa*. A phylogenetic analysis was conducted using a combined dataset consisting of large subunit ribosomal RNA (LSU), small subunit ribosomal RNA (SSU), internal transcribed spacer (ITS), beta-tubulin (tub2) and RNA polymerase II largest subunit 2 (rpb2) to infer their phylogenetic relationships.

## 2. Materials and Methods

### 2.1. Sampling, Isolation, and Morphological Examination

In 2023, samples of decaying culms of bamboo and submerged wood were collected from the Guangdong and Yunnan provinces in China. The specimens were placed in a zip-lock bag and taken to the laboratory for morphological examination. A stereomicroscope (Chongqing Optec Instrument Co., Ltd., Chongqing, China) was utilized to examine the colonies developed on the natural substrate. The micro fungal structures were photographed through a compound microscope (Nikon Eclipse Ni-U, Tokyo, Japan) fitted with a digital camera (Canon 750D, Tokyo, Japan). The measurement of fungal structures (conidiophores, conidiogenous cells and conidia) was conducted using the TaroSoft (R) Image Frame Work program v. 0.9.0.7. The single spore isolations were made following the method described by Senanayake et al. [21]. Herbarium specimens were deposited at the Mycological Herbarium of Zhongkai University of Agriculture and Engineering, Guangzhou, China (MHZU). The living cultures were deposited into the Zhongkai University of Agriculture and Engineering Culture Collection, Guangzhou, China (ZHKUCC). The novel species were registered in the databases of the Index Fungorum (http://www.indexfungorum.org/names/names.asp, accessed on 24 February 2024) [15] and Facesoffungi (http://www.facesoffungi.org, accessed on 24 February 2024) [22].

### 2.2. DNA Extraction, PCR Amplification, and Sequencing

The fungal mycelia were cultivated on a potato dextrose agar (PDA) medium at 28 °C for 30 days. A Maglso plant DNA isolation kit (Magen, Guangzhou, China) was employed to extract the genomic DNA following the instructions provided by the manufacture. DNA amplification was conducted using the polymerase chain reaction (PCR) technique, employing a 2720 Thermal Cycler instrument (Applied Biosystems, Foster City, CA, USA). LSU, ITS, SSU, tub2, and rpb2 were amplified using the primer pairs LR0R/LR5 [23], ITS5/ITS4 [24], NS1/NS4 [24], T1/BT2b [25,26], and fRPB2-5f/fRPB2-7cR [27], respectively. The amplifications were carried out in a 25 μL reaction volume containing 9.5 μL of ddH_2_O, 12.5 μL of 2 × FastTaq PCR Master Mix (Vazyme Co., Nanjing, China), 1 μL of DNA template, and 1 μL of each forward and reverse primer (10 μM). The PCR thermal cycles program for the amplification of LSU, ITS, SSU, and tub2 commenced with an initial denaturation step at 94 °C for 3 min, followed by 35 cycles consisting of denaturation at 94 °C for 30 s, annealing at 53 °C for 30 s, elongation at 72 °C for 60 s, and a final extension step at 72 °C for 10 min. For rpb2 amplification, the annealing temperature was adjusted to 56 °C. PCR products were checked on 1% agarose electrophoresis gels stained with Gel Red. The products with bright bands were transferred to Tianyi Huiyuan Biotechnology Co., Ltd. (Guangzhou, China) for sequencing reactions.

### 2.3. Phylogenetic Analyses

The quality of the sequences was initially evaluated using SeqMan v. 7.0.0 [28]. To evaluate the sequence similarities between the new collections and other species, the newly generated sequences were subjected to BLASTn searches utilizing the powerful Basic Local Alignment Search Tool (BLAST) (https://www.ncbi.nlm.nih.gov/, accessed on 10 October 2023). The reference sequences were obtained from the GenBank (Table 1) through BLASTn search results and relevant publications [4,7,12,29]. Each dataset of the LSU, ITS, SSU, tub2, and rpb2 was aligned using the online version of MAFFT v. 7.0362 (https://mafft.cbrc.jp/alignment/server/, accessed on 10 October 2023) [30] with default settings, and manually adjusted using BioEdit v. 7.0 [31]. The aligned datasets were merged using Mesquite v. 3.81 [32]. The Alignment Transformation Environment online program (https://sing.ei.uvigo.es/ALTER/, accessed on 10 October 2023) was used to convert the FASTA file to PHYLIP and NEXUS formats for phylogenetic analyses. The phylogenetic analyses were conducted using maximum likelihood (ML) [33] and Bayesian inference (BI) approaches [34].

The ML analysis was conducted using RAxML-HPC2 on XSEDE v. 8.2.8 [33] within the CIPRES Science Gateway platform [35], employing the GTR+I+G model of evolution. For BI analysis, the evolutionary model of nucleotide substitution analysis was selected independently for each locus using MrModeltest v2.3 [36]. The best-fit model for each gene selected by Akaike Information Criterion (AIC) is GTR+I+G. Markov Chain Monte Carlo sampling (MCMC) was used to evaluate Bayesian posterior probabilities (BYPP) [37]. MCMC sampling was run for 1,000,000 generations, and the trees were sampled every 100th generation. The first 25% of the trees that represented the burn-in phase were discarded, and the remaining 75% of the trees were used for calculating the posterior probabilities (PP) for the majority rule consensus tree. The resulting trees were viewed in FigTree v. 1.4.0 (Institute of Evolutionary Biology, University of Edinburgh, Edinburgh, UK) [38], and edited using Microsoft Office Power Point 2007 (Microsoft Corporation, Redmond, WA, USA).

## 3. Results

We base our new species following the principle of multiple evidence, as in the guidelines of Chethana et al. [39].

### 3.1. Phylogenetic Analyses

The combined sequence alignments comprised 98 taxa (Table 1) with *Amniculicola immersa* CBS 123083 and *Amniculicola parva* CBS 123092 as the outgroup taxa [7]. The dataset comprised 4514 characters including alignment gaps (LSU, ITS, SSU, tub2, and rpb2 sequence data). The RAxML analysis of the combined dataset yielded a best scoring tree with a final ML optimization likelihood value of -26757.620769. The matrix had 1662 distinct alignment patterns, with 50.02% undetermined characters or gaps. The estimated base frequencies were as follows: A = 0.243853, C = 0.250309, G = 0.276052, and T = 0.229786; substitution rates were AC = 2.252767, AG = 4.258052, AT = 1.766702, CG = 1.384778, CT = 8.963127, and GT = 1.000000; the gamma distribution shape parameter was α = 0.549203. Bayesian posterior probabilities from Bayesian inference analysis were assessed with a final average standard deviation of split frequencies = 0.010798. In the phylogenetic tree (Figure 1), our new collections ZHKUCC 24-0005 and ZHKUCC 24-0007 cluster within *Aquatisphaeria*, while ZHKUCC 24-0006 and ZHKUCC 24-0008 are nested within *Pseudotetraploa*. In addition, ZHKUCC 24-0009 is grouped within *Tetraploa*.

### 3.2. Taxonomy

#### 3.2.1. Identification of the New Collections

***Aquatisphaeria thailandica*** W.L. Li, D.F. Bao & Jian K. Liu, Phytotaxa 513(2): 122 (2021) (Figure 2)

Index Fungorum number: IF839208; Facesoffungi number: FoF 11751

*Saprobic* on decaying culms of bamboo. **Sexual morph:** undetermined. **Asexual morph:** hyphomycetous. *Colonies* on natural substratum effuse, scattered, dark brown to black. *Conidiophores* 42–80 × 4–5 μm ($\bar{x}$ = 64 × 4.3 μm, *n* = 15), macronematous, mononematous, erect, cylindrical, mostly straight, curved at base, with rounded or slightly tapering apex, unbranched, asymmetrically 3–6-septate, not constricted at septa, unevenly brown, darker in some cells, smooth-walled, thin-walled. *Conidiogenous cells* 3–5 × 2–4 μm, holoblastic, monoblastic, integrated, determinate, terminal, subcylindrical, trapezoidal, brown. *Conidia* 18–38 × 15–25 μm ($\bar{x}$ = 28 × 19 μm, *n* = 5), solitary, acrogenous, turbinate, dictyoseptate, muriform, composed of several angular cells, obscured because of dark pigmentation, brown, dark brown to black, occasionally olivaceous when young, smooth-walled, thin-walled, with a wedge-shaped, dark brown basal cell, 4–9 × 3–7 μm ($\bar{x}$ = 6 × 4 μm, *n* = 15), with four (occasionally two) apical appendages. *Appendages* 11–20 × 2.5–4 μm ($\bar{x}$ = 15 × 3.2 μm, *n* = 30), almost equal in length, subcylindrical, aseptate, unbranched, upward, brown, occasionally olivaceous, subhyaline at the apex. *Conidial secession* schizolytic.

**Culture characteristics:** The colonies on the PDA reach 50 mm in diam. after one month at 25 °C, circular, dark greyish brown in the middle, pale brown at the margin, with a dense aerial mycelium, velvety; from below, grey to dark brown in the middle, milky at the margin, with an entire edge.

**Material examined:** CHINA, Guangdong Province, Yangjiang City, on decaying culms of bamboo, 9 April 2023, H.J. Zhao, YG010 (MHZU 24-0005); living culture ZHKUCC 24-0005. GenBank numbers: LSU: PP336665, ITS: PP336657, SSU: PP336662, tub2: PP346803, tef1-α: PP346810.

**Notes:** The new collection ZHKUCC 24-0005 is very similar to *Aquatisphaeria thailandica* in all morphological characteristics, except for the smaller dimensions of the conidia (18–38 × 15–25 μm vs. 36–50 × 30–47 μm) and the shorter appendages (11–20 × 2.5–4 μm vs. mostly 19–29 × 2–3 μm) [5]. Regarding the DNA sequence data comparison, there is a discrepancy of 0.8% (4 out of 505), 0.1% (1 out of 841) and 0.1% (1 out of 1031) in nucleotide variations within the ITS, LSU, and SSU genes, respectively, between ZHKUCC 24-0005 and *A. thailandica* MFLUCC 21-0025. No other protein coding genes can be compared. Based on multi-locus phylogenetic analysis and morphological comparison, the new collection ZHKUCC 24-0005 is identified as *A. thailandica*. This study provides a new geographical record of *A. thailandica* in China, as well as a new terrestrial habitat record for this species. In addition, this study expands the range of conidial dimensions of *A. thailandica* to 18–50 × 15–47 μm. The limitations of the specimen materials need to be clarified, as our data are based on a small sample size of only five conidia obtained from the substrate.

***Aquatisphaeria bambusae*** H.J. Zhao, K.D. Hyde & W. Dong, sp. nov. (Figure 3)

Index Fungorum number: IF 901707; Facesoffungi number: FoF 15535

**Etymology:** refers to the bamboo host of the holotype.

**Holotype:** MHZU 24-0007

*Saprobic* on decaying culms of bamboo. **Sexual morph:** undetermined. **Asexual morph:** hyphomycetous. *Colonies* on natural substratum effuse, scattered, dark brown to black. *Conidiophores* 42–93 × 4–7 μm ($\bar{x}$ = 73 × 5.5 μm, *n* = 15), macronematous, mononematous, erect, cylindrical, straight, with a rounded apex, unbranched, 3–7-septate, not constricted at the septa, dark brown to black, smooth-walled, thin-walled. *Conidiogenous cells* 10.5 × 3 μm, holoblastic, monoblastic, integrated, determinate, terminal, subcylindrical, dark brown to black. *Conidia* 26–57 × 22–35 μm ($\bar{x}$ = 36.5 × 29.5 μm, *n* = 30), solitary, acrogenous, subglobose, turbinate, dictyoseptate, muriform, clearly septate and becoming aseptate, brown when young, dark brown at maturity, rough-walled, verrucose, thin-walled, with a subcylindrical, wedge-shaped, dark brown basal cell, which is 3.5–7.5 × 3–5.5 μm ($\bar{x}$ = 5.0 × 4.1 μm, *n* = 15), with two to five (mostly four) appendages. *Appendages* 15–35 × 2–7 μm ($\bar{x}$ = 25 × 3.6 μm, *n* = 30), almost equal in length, cylindrical, aseptate, unbranched, upward, olivaceous brown to dark brown, subhyaline at the apex. *Conidial secession* schizolytic.

**Culture characteristics:** The colonies on the PDA reach 22 mm in diam. after two weeks at 25 °C, circular, pale grey, grey at the margin, with a dense aerial mycelium, velvety; from below, pale brown, with a white entire edge.

**Material examined:** CHINA, Guangdong Province, Yangjiang City, on decaying culms of bamboo, 9 April 2023, H.J. Zhao, YG048 (MHZU 24-0007, holotype); ex-type culture ZHKUCC 24-0007. GenBank numbers: LSU: PP336667, ITS: PP336659, SSU: PP336663, tub2: PP346805, tef1-α: PP346812.

**Notes:** In our multi-locus phylogenetic analysis, *Aquatisphaeria bambusae* forms a sister branch with *A. thailandica* (Figure 1). Morphologically, *A. bambusae* can be distinguished from *A. thailandica* by differences in conidial septation and color. *Aquatisphaeria thailandica* has distinct septa that are prominently pigmented at maturity, whereas the septa of *A. bambusae* become indiscernible and display a lighter color compared to those of *A. thailandica* [5]. In addition, *A. bambusae* has conidia that are thinner (26–57 × 22–35 μm) compared to those of *A. thailandica* (36–50 × 30–47 μm). *Aquatisphaeria bambusae* possesses slightly longer appendages (15–35 × 2–7 μm) in comparison to *A. thailandica* (19–29 × 2–3 μm). Regarding the DNA sequence data comparison, there is a discrepancy of 3% (15 out of 503), 0.7% (6 out of 849), and 0.1% (1 out of 1021) in nucleotide variations within the ITS, LSU, and SSU genes, respectively, between *A. bambusae* ZHKUCC 24-0007 and *A. thailandica* MFLUCC 21-0025. Based on multi-locus phylogenetic analysis and morphological comparison [39], *A. bambusae* is identified as a new species within *Aquatisphaeria*.

***Pseudotetraploa yangjiangensis*** H.J. Zhao, K.D. Hyde & W. Dong, sp. nov. (Figure 4)

Index Fungorum number: IF 901708; Facesoffungi number: FoF 11536

**Etymology:** refers to Yangjiang City, from where the holotype was collected.

**Holotype:** MHZU 24-0008

*Saprobic* on decaying culms of *Phyllostachys edulis.*
**Sexual morph:** undetermined. **Asexual morph:** hyphomycetous. *Colonies* on natural substratum effuse, gregarious or scattered, black. *Conidiophores* reduced to conidiogenous cells. *Conidiogenous cells* holoblastic, monoblastic, integrated. *Conidia* 17–50 × 15–32 μm ($\bar{x}$ = 36 × 20 μm, *n* = 20), solitary, pyriform, composed of 3–5 columns of brown to dark brown cells, pseudoseptate, rough-walled, verrucose, thin-walled, with 1–5 apical appendages, with the middle two appendages mostly being the longest and straight, the other appendages being short and slightly curved. *Appendages* 10–95 × 3.3–7.6 μm ($\bar{x}$ = 42 × 4.7 μm, *n* = 30), subcylindrical, 1–4(–5)-septate, unbranched, upward, brown, subhyaline at the apex. *Conidial secession* schizolytic.

**Culture characteristics:** The colonies on the PDA reach 50 mm in diam. after five weeks at 25 °C, circular, dry, white, with dense mycelium; from below, dark brown in the middle, pale brown at the margin, with an entire edge.

**Material examined:** CHINA, Guangdong Province, Yangjiang City, on decaying culms of *Phyllostachys edulis*, 9 April 2023, H.J. Zhao, YG092 (MHZU 24-0008, holotype); ex-type living culture, ZHKUCC 24-0008. GenBank numbers: LSU: PP336668, ITS: PP336660, tub2: PP346806, rpb2: PP346809.

**Notes:** In our multi-locus phylogenetic analysis, *Pseudotetraploa yangjiangensis* has close affinities with *P. rajmachiensis* (Figure 1). Morphologically, *P. yangjiangensis* can be distinguished from *P. rajmachiensis* by having 3–5 columnar conidia, whereas *P. rajmachiensis* possesses only 2–3 columnar conidia [18]. In addition, the conidia of *P. yangjiangensis* are wider (17–50 × 15–32 μm) compared to those of *P. rajmachiensis* (27.6–43 × 16–23.5 µm). *Pseudotetraploa yangjiangensis* possesses slightly longer appendages with fewer septa (10–94 μm, 1–4(–5)-septate) in comparison of *P. rajmachiensis* (16.5–79 μm, 4–10-septate). Regarding the DNA sequence data comparison, there is a discrepancy of 1.9% (16 out of 861, 1 gap), 1.5% (12 out of 824, 1 gap) and 4.0% (14 out of 354, 1 gap) in nucleotide variations within the LSU, ITS, and tub2 genes, respectively, between *P. yangjiangensis* ZHKUCC 24-0008 and *P. rajmachiensis* NFCCI 4618. Based on multi-locus phylogenetic analysis and morphological comparison [39], *P. yangjiangensis* is identified as a new species within *Pseudotetraploa*.

***Pseudotetraploa phyllostachydis*** H.J. Zhao, K.D. Hyde & W. Dong, sp. nov. (Figure 5)

Index Fungorum number: IF 901709; Facesoffungi number: FoF 15537

**Etymology:** refers to host name of the holotype.

**Holotype:** MHZU 24-0006

*Saprobic* on decaying culms of *Phyllostachys edulis.*
**Sexual morph:** undetermined. **Asexual morph:** hyphomycetous. *Colonies* on natural substratum effuse, scattered or gregarious, black. *Conidiophores* reduced to conidiogenous cells. *Conidiogenous cells* holoblastic, monoblastic, integrated. Conidia 17–30 × 10–20 μm ($\bar{x}$ = 25 × 14.7 μm, *n* = 30), solitary, ovoid, brown to dark brown, sometimes olivaceous brown in the upper portion, composed of two to four columns of appressed, brown to dark brown cells, pseudoseptate, rough-walled, verrucose, thin-walled, with 1–4 apical appendages, with the middle appendage being the longest and straight, the other appendages being short and distinctly curved. *Appendages* 9–75 × 2.8–8 μm ($\bar{x}$ = 35.5 × 4.3 μm, *n* = 30), subcylindrical, 0–4-septate, unbranched, upward, brown to dark brown, subhyaline at the apex. *Conidial secession* schizolytic.

**Culture characteristics:** The colonies on the PDA reach 50 mm in diam. after five weeks at 25 °C, circular, dry, with dense mycelium, off-white; from below, dark brown in the middle, white to pale brown outwards, white at the margin, with a filiform edge.

**Material examined:** CHINA, Guangdong Province, Yangchun City, on decaying culms of *Phyllostachys edulis*, 9 April 2023, H.J. Zhao, YG029 (MHZU 24-0006, holotype); ex-type culture ZHKUCC 24-0006. GenBank numbers: LSU: PP336666, ITS: PP336658, tub2: PP346804, rpb2: PP346808, tef1-α: PP346811.

**Notes:** In our multi-locus phylogenetic analysis, *Pseudotetraploa phyllostachydis* forms a basal branch to *P. rajmachiensis* and *P. yangjiangensis* (Figure 1). *Pseudotetraploa phyllostachydis* can be easily distinguished from *P. rajmachiensis* and *P. yangjiangensis* by the conidia that only possess a single longest middle appendage compared to the predominantly two longest middle appendages found in the latter two species [18]. In addition, *P. phyllostachydis* exhibits shorter conidia (17–30 × 10–20 μm) compared to *P. rajmachiensis* (27.6–43 × 16–23.5 µm), and smaller conidia than *P. yangjiangensis* (17–50 × 15–32 μm). The longest appendages of *P. phyllostachydis* measure 75 µm, whereas those of *P. yangjiangensis* measure 94 μm. Regarding the DNA sequence data comparison, there is a discrepancy of 3.8% (33 out of 871, 1 gap), 7.9% (43 out of 541, 12 gaps), and 8.9% (31 out of 350, 4 gaps) in nucleotide variations within the LSU, ITS, and tub2 genes, respectively, between *P. phyllostachydis* ZHKUCC 24-0006 and *P. rajmachiensis* NFCCI 4618; there is a discrepancy of 3.1% (27 out of 865), 8.6% (47 out of 543, 10 gaps), and 7.7% (42 out of 547, 9 gaps) in nucleotide variations within the LSU, ITS, and tub2 genes, respectively, between *P. phyllostachydis* ZHKUCC 24-0006 and *P. yangjiangensis* ZHKUCC 24-0008. Based on multi-locus phylogenetic analysis and morphological comparison [39], *P. phyllostachydis* is identified as a new species within *Pseudotetraploa*.

***Tetraploa submersa*** H.J. Zhao, G.N. Wang, K.D. Hyde & W. Dong, sp. nov. (Figure 6)

Index Fungorum number: IF 901710; Facesoffungi number: FoF 15538

**Etymology:** refers to submerged habitats of the holotype.

**Holotype:** MHZU 24-0009

*Saprobic* on submerged wood in freshwater habitats. **Sexual morph:** undetermined. **Asexual morph:** hyphomycetous. *Colonies* on natural substratum effuse, scattered or gregarious, black. *Conidiophores* reduced to conidiogenous cells. *Conidiogenous cells* up to 6 μm long, holoblastic, monoblastic, integrated, subcylindrical, pale brown. *Conidia* 30–68 × 21–53 μm ($\bar{x}$ = 38 × 29.5 μm, *n* = 30), solitary, obovoid, cuneiform, subcylindrical or subspherical, often divergent and digitate, brown to dark brown, composed of 3–5 columns of brown to dark brown cells, 3–5-septate in each column, smooth-walled, thin-walled, with 3–5 apical appendages. *Appendages* 8–53 μm long ($\bar{x}$ = 25 μm, *n* = 30), 2–6 µm wide at the apex, 3–10 µm wide at the base, almost equal in length but with one to two appendages often continuing to grow, forming a subcylindrical, paler cell that is measured as 10–40 (–60) μm long, subcylindrical, slightly wider at the base, 1–3-septate, unbranched, upward, pale brown to brown. *Conidial secession* schizolytic.

**Culture characteristics:** The colonies on the PDA reach 52 mm in diam. after five weeks at 25 °C, circular, cottony, flat, pale brown; from below, dark brown, with an undulate, pale brown edge.

**Material examined:** CHINA, Yunnan Province, Dehong Dai Jingpo Autonomous Prefecture, Mang City, on submerged wood in freshwater habitats, 9 March 2023, G.N. Wang, YN1 (MHZU 24-0009, holotype); ex-type culture ZHKUCC 24-0009. GenBank numbers: LSU: PP336669, ITS: PP336661, SSU: PP336664, tub2: PP346807, tef1-α: PP346813.

**Notes:** In our multi-locus phylogenetic analysis, *Tetraploa submersa* clusters with *T. pseudoaristata*, *T. puzheheiensis,* and two unnamed strains, CY 112 and CBS 996.70 (Figure 1). *Tetraploa submersa* can be easily distinguished from *T. puzheheiensis* in having cuneiform, brown to dark brown, smooth-walled conidia that have conidial arms which often become divergent and digitate at maturity. In contrast, *T. puzheheiensis* exhibits short cylindrical, almost black, verrucose conidia with closed conidial arms [4]. In addition, the appendages of *T. submersa* exhibit greater length (8–53 μm), with one to two of them often continuing to grow and forming a subcylindrical, paler cell that measures 10–40(–60) μm in length. Conversely, the appendages of *T. puzheheiensis* are shorter (3–27 µm) and do not undergo further growth at maturity. *Tetraploa pseudoaristata* can also be easily distinguished from *T. submersa* in having short cylindrical, verrucose conidia with almost closed conidial arms and with quite long appendages that measure 23–107.5 μm [18]. The three species can also be distinguished by conidial size, with *T. submersa* measuring 30–68 × 21–53 μm, *T. puzheheiensis* measuring 24–35 × 15–24.5 μm, and *T. pseudoaristata* measuring 22–31 × 15–20 µm [4,18]. The strains CY 112 and CBS 996.70, bearing the name *T. scheueri* and *T. aristata*, were identified as doubtful strains in Dong et al. [4], and were provisionally classified as *Tetraploa* spp. Nevertheless, the comparison of *T. scheueri* and *T. aristata* with *T. submersa* indicate that they are significantly distinct species (morphology see [1,4,40]).

Regarding the DNA sequence data comparison, there is a discrepancy of 0.5% (4 out of 865), 2.4% (13 out of 539, 3 gaps) and 4.8% (20 out of 413) in nucleotide variations within the LSU, ITS, and tub2 genes, respectively, between *T. submersa* ZHKUCC 24-0009 and *T. pseudoaristata* NFCCI 4624; there is a discrepancy of 0.5% (4 out of 852, 1 gap) and 2.0% (11 out of 553, 2 gaps) in nucleotide variations within the LSU and ITS genes, respectively, between *T. submersa* ZHKUCC 24-0009 and *T. puzheheiensis* KUMCC 20-0151. To the best of our knowledge, *T. submersa* does not correspond to any known species within the genus. Therefore, it is introduced as a novel species of *Tetraploa*.

#### 3.2.2. Overview of the Taxa Belonging to Tetraplosphaeriaceae

A compilation of the Tetraplosphaeriaceae species isolated from bamboo (marked with “*****”) or freshwater (marked with “**#**”) is provided below.

***Aquatisphaeria*** W.L. Li, N.G. Liu & Jian K. Liu, Phytotaxa 513(2): 122 (2021)

****Aquatisphaeria bambusae*** H.J. Zhao & W. Dong, J. Fungi 10, 5: 319 (2024)—on decaying culms of an unidentified bamboo: China [this study]

**#**Aquatisphaeria thailandica*** W.L. Li, D.F. Bao & Jian K. Liu, Phytotaxa 513(2): 122 (2021)—on decaying wood submerged in freshwater habitats: Thailand [5]; on decaying culms of bamboo: China [this study]

**Notes:** *Aquatisphaeria* was established by Li et al. [5] with *A. thailandica* as the type species. Within the genus, *A. thailandica* and *A. bambusae* have been documented, both of which were isolated from decaying bamboo [5] and this study. In particular, *A. thailandica* also occurs in freshwater habitats. The proportion of the species within *Aquatisphaeria* that were isolated from bamboo substrates and/or freshwater habitats is 100% (2/2).

***Byssolophis*** Clem., Gen. fung., Edn 2 (Minneapolis): 83 (1931)

**Notes:** *Byssolophis ampla*, *B. byssiseda*, and *B. sphaerioides* have been documented within the genus [15]. No species have been reported to be found from bamboo substrates or freshwater habitats.

***Ernakulamia*** Subram., Kavaka 22/23: 67 (1996) [1994]

**#*Ernakulamia cochinensis*** (Subram.) Subram., Kavaka 22/23: 67 (1996) [1994]—on submerged wood in a stream: Thailand [4]

**Notes:** *Ernakulamia* consists of four species, as documented in Species Fungorum (https://www.speciesfungorum.org/, accessed on 15 February 2024). Bambusicolous species remains to be discovered within the genus; however, *E. cochinensis* has been found from freshwater habitats [4]. The proportion of the species within *Ernakulamia* that were isolated from bamboo substrates and/or freshwater habitats is 25% (1/4).

***Polyplosphaeria*** Kaz. Tanaka & K. Hiray., Stud. Mycol. 64: 192 (2009)

****Polyplosphaeria fusca*** Kaz. Tanaka & K. Hiray., Stud. Mycol. 64: 193 (2009)—on decaying culms of *Chimonobambusa marmorea*, *Phyllostachys bambusoides*, *Pleioblastus chino*, and *Sasa kurilensis*: Honshu, Japan [1]; on decaying stems of an unidentified bamboo: Yunnan, China [41]; on decaying culms of an unidentified bamboo: Guizhou, China [10]

****Polyplosphaeria thailandica*** C.G. Lin, Yong Wang bis & K.D. Hyde, Fungal Diversity 78: 55 (2016)—on decaying culms of an unidentified bamboo: Thailand [42]

**Notes:** *Polyplosphaeria* was established by Tanaka et al. [1] with *P. fusca* as the type species. Currently, five species are accepted within the genus. Among them, two species, viz., *P. fusca* and *P. thailandica*, were isolated from decaying bamboo [1,42]. No species have been found from freshwater habitats. The proportion of the species within *Polyplosphaeria* that were isolated from bamboo substrates and/or freshwater habitats is 29% (2/7).

***Pseudopolyplosphaeria*** J.F. Zhang, Y.Y. Chen & Jian K. Liu, Fungal Divers. 122: 59 (2023)

****Pseudopolyplosphaeria guizhouensis*** J.F. Zhang, Y.Y. Chen & Jian K. Liu, Fungal Diversity 122: 61 (2023)—on decaying culms of an unidentified bamboo: Guizhou, China [10]

**Notes:** *Pseudopolyplosphaeria* was established by Zhang et al. [10] with *P. guizhouensis* as the type species. Currently, only one species, *P. guizhouensis,* has been documented within the genus, which was isolated from the decaying culms of an unidentified bamboo. No species have been found from freshwater habitats. The proportion of the species within *Pseudopolyplosphaeria* that were isolated from bamboo substrates and/or freshwater habitats is 100% (1/1).

***Pseudotetraploa*** Kaz. Tanaka & K. Hiray., Stud. Mycol. 64: 193 (2009)

****Pseudotetraploa bambusicola*** X.D. Yu, S.N. Zhang & Jian K. Liu, Journal of Fungi 8 (7, no. 720): 16 (2022)—on decaying stems of Bambusoideae: Sichuan, China [6]

****Pseudotetraploa curviappendiculata*** (Sat. Hatak., Kaz. Tanaka & Y. Harada) Kaz. Tanaka & K. Hiray., Stud. Mycol. 64: 195 (2009)—on decaying culms of *Sasa kurilensis*: Honshu, Japan [17]

****Pseudotetraploa javanica*** (Rifai, Zainuddin & Cholil) Kaz. Tanaka & K. Hiray., Stud. Mycol. 64: 195 (2009)—on decaying culms of *Bambusa glaucescens*: Jawa (Rifai et al. 1988); on decaying culms of *Pleioblastus chino*: Japan [1]

****Pseudotetraploa longissima*** (Sat. Hatak., Kaz. Tanaka & Y. Harada) Kaz. Tanaka & K. Hiray., Stud. Mycol. 64: 195 (2009)—on decaying culms of *Pleioblastus chino*: Honshu, Japan [1]

****Pseudotetraploa phyllostachydis*** H.J. Zhao, K.D. Hyde & W. Dong, J. Fungi 10(5): 319 (2024)—on decaying culms of *Phyllostachys edulis*: Guangdong, China [this study]

****Pseudotetraploa rajmachiensis*** Rajeshkumar, K.D. Hyde & Wijayaw., Fungal Divers. 100: 116 (2020)—on decaying culms of *Dendrocalamus stocksii*: Maharashtra, India [18]

****Pseudotetraploa yangjiangensis*** H.J. Zhao, K.D. Hyde & W. Dong, J. Fungi 10(5): (2024)—on decaying culms of *Phyllostachys edulis*: Guangdong, China [this study]

**#**Pseudotetraploa yunnanensis*** X. Tang, Jayaward., R. Jeewon & J.C. Kang, MycoKeys 100: 187 (2023)—on submerged bamboo in freshwater habitats: Yunnan, China [7]

**Notes:** *Pseudotetraploa* was established by Tanaka et al. [1] with *P. curviappendiculata* as the type species. All species were isolated from bamboo substrates in previous studies [1,6,7,17,18] and in this study. In particular, *P. yunnanensis* has also been found in freshwater habitats [7]. The proportion of the species within *Pseudotetraploa* that were isolated from bamboo substrates and/or freshwater habitats is 100% (8/8).

***Quadricrura*** Kaz. Tanaka, K. Hiray. & Sat. Hatak., Stud. Mycol. 64: 196 (2009)

****Quadricrura meridionalis*** Kaz. Tanaka & K. Hiray., Stud. Mycol. 64: 197 (2009)—on culms of an unidentified bamboo: Nansei-Shoto [1]

****Quadricrura septentrionalis*** Kaz. Tanaka, K. Hiray. & Sat. Hatak., Stud. Mycol. 64: 198 (2009)—on culms of *Sasa kurilensis*: Honshu, Japan [1]

**Notes:** *Quadricrura* was established by Tanaka et al. [1] with *Q. septentrionalis* as the type species. Three species have been documented, two of which, viz., *Q. meridionalis* and *Q. septentrionalis*, were isolated from decaying bamboo [1]. No species have been found from freshwater habitats. The proportion of the species within *Quadricrura* that were isolated from bamboo substrates and/or freshwater habitats is 67% (2/3).

***Shrungabeeja*** V.G. Rao & K.A. Reddy, Indian J. Bot. 4(1): 109 (1981)

**#*Shrungabeeja aquatica*** W. Dong, G.N. Wang & H. Zhang, Fungal Divers. 105: 498 (2020)—on submerged wood in freshwater habitats: Thailand [4]

**#*Shrungabeeja fluviatilis*** J. Yang, Jian K. Liu & K.D. Hyde, Fungal Divers. 119: 74 (2023)—on decaying twigs submerged in freshwater habitats: Guizhou, China [8]

****Shrungabeeja longiappendiculata*** Sommai, Pinruan, Nuankaew & Suetrong, Fungal Divers. 75: 124 (2015)—on decaying culms of *Bambusa*: Thailand [3]

****Shrungabeeja piepenbringiana*** R. Kirschner, Sydowia 69: 155 (2017)—on decaying detached twigs of *Chusquea longifolia*: Panama [43]

**#**Shrungabeeja vadirajensis*** V.G. Rao & K.A. Reddy, Indian J. Bot. 4(1): 113 (1981)—on decaying stems of *Bambusa*: Karnataka, India [44]; on submerged wood in a stream: Thailand [4]

**Notes:** *Shrungabeeja* consists of seven species, as documented in Species Fungorum (https://www.speciesfungorum.org/, accessed on 15 February 2024). Among them, five species have been documented from decaying bamboo or freshwater habitats. In particular, *S. vadirajensis* was found both on decaying bamboo and in freshwater habitats [4,44]. The proportion of the species within *Shrungabeeja* that were isolated from bamboo substrates and/or freshwater habitats is 71% (5/7).

***Tetraploa*** Berk. & Broome, Ann. Mag. nat. Hist., Ser. 2 5: 459 (1850)

**#*Tetraploa abortiva*** Aramb. & Cabello, Mycotaxon 30: 266 (1987)—on submerged wood in freshwater habitats: Argentina [45]

**#*Tetraploa aquatica*** W.L. Li & H.Y. Su, in Li, Bao, Bhat & Su, Phytotaxa 459(2): 184 (2020)—on decaying wood submerged in freshwater habitats: Yunnan, China [13]

****Tetraploa bambusae*** Phookamsak & H.B. Jiang, J. Fungi 8 (no. 630): 25 (2022)—on decaying twigs of Bambusoideae: Yunnan, China [12]

****Tetraploa circinata*** J. Pratibha & Bhat, Mycotaxon 105: 423 (2008)—on dead, decaying twigs of an unidentified bamboo: Maharashtra, India [12]

****Tetraploa nagasakiensis*** (Kaz. Tanaka & K. Hiray.) Kaz. Tanaka & K. Hiray., Fungal Divers. 63: 253 (2013)—on culms of an unidentified bamboo: Kyushu, Japan [1]; on decaying branches of an unidentified bamboo: Yunnan, China [46]

****Tetraploa opaca*** G.Z. Zhao, Nova Hedwigia 88(1-2): 223 (2009)—on decaying culms of an unidentified bamboo [47]

**#*Tetraploa puzheheiensis*** W. Dong, H. Yang & H. Zhang, Fungal Divers. 105: 501 (2020)—on submerged wood in freshwater habitats: Yunnan, China [4]

****Tetraploa sasicola*** (Kaz. Tanaka & K. Hiray.) Kaz. Tanaka & K. Hiray., Fungal Divers. 63: 253 (2013)—on culms of *Sasa senanensis*: Hokkaido, Japan [1]

**#*Tetraploa submersa*** H.J. Zhao, G.N. Wang, K.D. Hyde & W. Dong, J. Fungi 10(5): 319 (2024)—on submerged wood in freshwater habitats: Yunnan, China [this study]

**#*Tetraploa thailandica*** D.F. Bao, H.Y. Su, K.D. Hyde & Z.L. Luo, J. Fungi 7 (no. 669): 4 (2021)—on submerged wood in freshwater habitats: Thailand [48]

****Tetraploa yakushimensis*** (Kaz. Tanaka, K. Hiray. & Hosoya) Kaz. Tanaka & K. Hiray., Fungal Divers. 63: 253 (2013)—on culms of *Arundo donax*: Kagoshima, Japan [1]

**#*Tetraploa yunnanensis*** W. Dong, H. Yang & H. Zhang, Fungal Divers. 105: 502 (2020)—on submerged wood in freshwater habitats: Yunnan, China [4], Thailand [4]

**Notes:** *Tetraploa* was established by Berkeley and Broome [14] with *T. aristata* as the type species. A total of 39 species have been documented, with 11 of them being isolated from bamboo substrates and freshwater habitats. *Tetraploa bambusae*, *T. circinata*, *T. nagasakiensis*, *T. opaca*, *T. sasicola,* and *T. yakushimensis* were exclusively isolated from decaying bamboo (see records under the notes on the species), while *T. abortiva*, *T. aquatica*, *T. submersa*, *T. puzheheiensis*, *T. thailandica,* and *T. yunnanensis* have been exclusively isolated from freshwater habitats (see records under the notes on the species). The proportion of the species within *Tetraploa* that were isolated from bamboo substrates and/or freshwater habitats is 31% (12/39).

***Triplosphaeria*** Kaz. Tanaka & K. Hiray., Stud. Mycol. 64: 185 (2009)

**#**Triplosphaeria acuta*** Kaz. Tanaka & K. Hiray., Stud. Mycol. 64: 186 (2009)—on culms of an unidentified bamboo submerged in a stream: Hokkaido, Japan [1]

****Triplosphaeria cylindrica*** Kaz. Tanaka & K. Hiray., Stud. Mycol. 64: 188 (2009)—on culms of Sasa kurilensis, Japan [1]

**#*Triplosphaeria guizhouensis*** L.L. Liu & Z.Y. Liu, Phytotaxa 603(2): 177 (2023)—on decaying wood submerged in a stream: Guizhou, China [49]

****Triplosphaeria maxima*** Kaz. Tanaka & K. Hiray., Stud. Mycol. 64: 188 (2009)—on culms of *Sasa kurilensis*: Honshu, Japan [1]

****Triplosphaeria yezoensis*** (I. Hino & Katum.) Kaz. Tanaka & K. Hiray., Stud. Mycol. 64: 188 (2009)—on decaying culms of *Sasa paniculata*: Japan [1]

**Notes:** *Triplosphaeria* was established by Tanaka et al. [1] with *T. maxima* as the type species. All species have been isolated from decaying bamboo or freshwater habitats. Among them, *T. cylindrica*, *T. maxima*, and *T. yezoensis* were exclusively isolated from decaying bamboo [1], while *T. guizhouensis* was exclusively isolated from freshwater habitats [49]. In particular, *T. acuta* can inhabit both decaying bamboo and submerged wood in freshwater habitats [1]. The proportion of the species within *Triplosphaeria* that were isolated from bamboo substrates and/or freshwater habitats is 100% (5/5).

## 4. Discussion

Bambusicolous fungi are a diverse group associated with various bamboo substrates, including leaves, culms, branches, rhizomes, and roots. The extensive research from articles and monographs has confirmed the global distribution and high fungal diversity of bambusicolous fungi [1,7,50,51,52]. Freshwater fungi is another group that partially or completely resides in freshwater habitats, with numerous novel species having been discovered over the past decades [4,8,20,53,54,55,56,57,58,59]. The most common fungi in the two groups are generally clustered into distinct families. Hyde et al. [50] and Dai et al. [51] proposed that the common families of fungi on bamboo are Hypocreaceae, Phyllachoraceae and Xylariaceae, Bambusicolaceae, Tetraplosphaeriaceae, and Roussoellaceae. Dong et al. [4] summarized that freshwater fungi predominantly reside in Aliquandostipitaceae, Dictyosporiaceae, Morosphaeriaceae, Tetraplosphaeriaceae, and Tubeufiaceae. It is likely that the species within Tetraplosphaeriaceae exhibit a preference for inhabiting both bamboo substrates and freshwater habitats. To investigate this assumption, our study presents a compilation of Tetraplosphaeriaceae species that have been isolated from bamboo substrates and freshwater habitats, accompanied by notes for each genus explaining the isolation source for those particular species. Interestingly, we found that the species within the two genera, viz., *Pseudotetraploa* and *Triplosphaeria*, exclusively inhabit bamboo substrates and freshwater habitats, indicating their strong preference for these specific environments. The species identified within *Aquatisphaeria* and *Pseudopolyplosphaeria* have also been isolated from such environments; however, due to the limited sampling size thus far, it remains difficult to determine their definitive environmental preferences. The taxa within *Ernakulamia*, *Polyplosphaeria*, and *Tetraploa*, on the other hand, exhibit a relatively low percentage of species that inhabit such environments at 25%, 29%, and 31%, respectively. The species within *Quadricrura* and *Shrungabeeja* exhibit a moderate percentage at 67% and 71%, respectively. Four species, viz., *Aquatisphaeria thailandica*, *Pseudotetraploa yunnanensis*, *Shrungabeeja vadirajensis*, and *Triplosphaeria acuta*, are capable of thriving in both environments. In total, Tetraplosphaeriaceae comprises 48% (38/79) of species that have been isolated from bamboo substrates and freshwater habitats. In other words, 41 species in this family were isolated from other substrates or habitats, while 27 of them (66%) belong to the *Tetraploa*. *Tetraploa* species are frequently documented from herbaceous plants [1,60,61,62,63]. This prompts us to question whether the majority of species within *Tetraploa* have experienced migration from bamboo substrates or freshwater habitats to other herbaceous plants in response to significant environmental pressures. Numerous matters are still pending in the field of freshwater fungal biology.

The taxonomic investigations of microfungi have been conducted by various researchers, with an emphasis on their respective habitats, such as marine habitats [64], freshwater habitats [56], or an emphasis on their hosts, such as mangrove fungi [65], teak fungi [66], grass fungi [67], and entomopathogenic fungi [68]. Alternatively, researchers may focus on particular fungal groups, such as annulatascaceae-like taxa [57,69], tubeufia-like taxa [55], pestalotiopsis-like taxa [70,71]. However, the taxonomic studies on fungal groups, simultaneously providing valuable insights into their hosts and habitats, have been rarely investigated. The compilation presented in this study, along with our new findings, have suggested that the Tetraplosphaeriaceae is likely to exhibit a high degree of host and habitat specificity. This can be clearly observed from examples of *Aquatisphaeria*, *Pseudotetraploa*, *Quadricrura*, *Shrungabeeja,* and *Triplosphaeria*. The species within *Tetraploa* have not exhibited a distinct preference for bamboo substrates and freshwater habitats. Due to the rich fungal diversity and wide distribution of species in bamboo substrates and freshwater habitats, we believe that numerous novel species and genera will be discovered within Tetraplosphaeriaceae through extensive investigations into fungi from these specific substrates and habitats. In this study, five new collections were collected from decaying bamboo substrates in terrestrial habitats and submerged wood in freshwater habitats. Morphological and phylogenetic analyses have identified four novel species, viz., *Aquatisphaeria bambusae* sp. nov., *Pseudotetraploa yangchunensis* sp. nov., *Pseudotetraploa yangjiangensis* sp. nov., and *Tetraploa submersa* sp. nov., as well as a previously documented species *Aquatisphaeria thailandica* within Tetraplosphaeriaceae. *Aquatisphaeria thailandica* was initially discovered from a freshwater habitat on submerged wood in Thailand, and we first report finding this species in a terrestrial habitat on decaying bamboo substrate in China.

The phylogenetic relationships between genera in Tetraplosphaeriaceae have been extensively investigated utilizing DNA sequence data [1,5,6,7,10]. However, the classification of several species, such as *Polyplosphaeria thailandica*, remains uncertain. *Polyplosphaeria thailandica* was introduced within *Polyplosphaeria* based on morphological and phylogenetic analysis [42]. However, subsequent phylogeny revealed that *P. thailandica* does not cluster with other species in the genus *Polyplosphaeria* but instead shows closer phylogenetic affinities with *Aquatisphaeria*. The BLASTn searches on *P. thailandica* using the LSU sequence reveals *P. thailandica* to have the closest relationships to *Aquatisphaeria thailandica* (98.44%), followed by *Quadricrura bicornis* (97.58%), and *Triplosphaeria acuta* (97.58%). The BLASTn searches of the ITS sequence reveals *P. thailandica* to have the closest relationships to *Hermatomyces sphaericus* (94.68%), followed by *Aquatisphaeria thailandica* (94.13%), and *Shrungabeeja fluviatilis* (94.09%). The results from both LSU and ITS sequence Blast analyses showed that *P. thailandica* exhibited phylogenetic affinities with *Aquatisphaeria*. However, based on its morphology, *P. thailandica* should be classified under *Polyplosphaeria* rather than *Aquatisphaeria* [1,5,42]. Considering the contradiction between the morphology and phylogeny, the identification of *P. thailandica* is pending, to be resolved with new collections.

## Figures and Tables

**Figure 1 jof-10-00319-f001:**
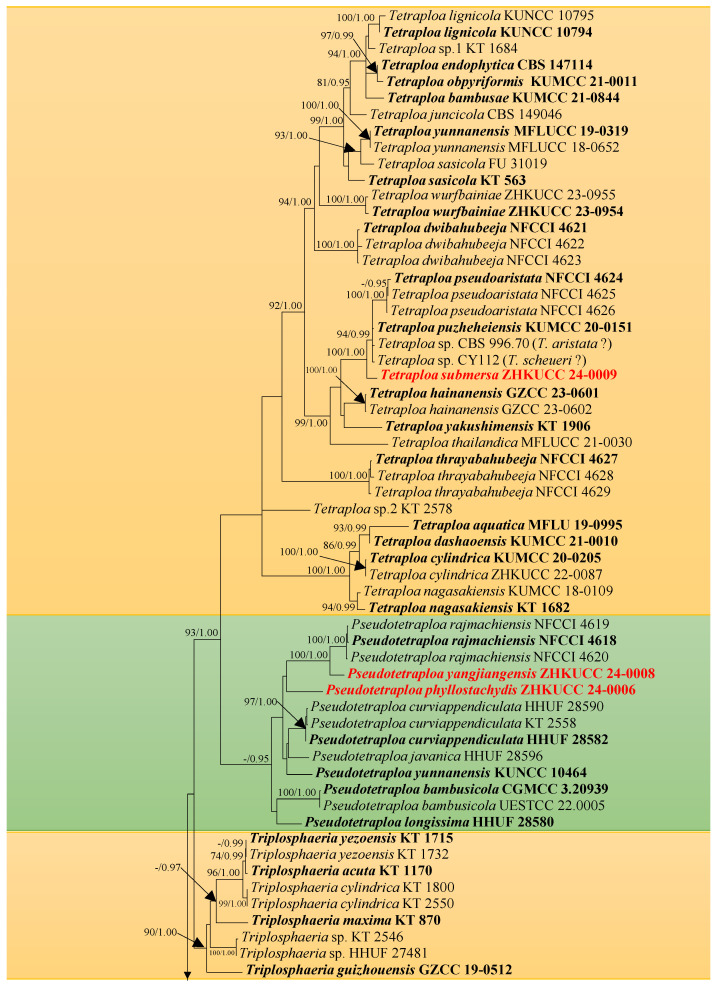
Phylogenetic tree generated from maximum likelihood (ML) analysis based on combined LSU, ITS, SSU, tub2, and rpb2 sequence data. Bootstrap support values for ML equal to greater than 70%, and posterior probabilities equal to greater than 0.95 are given above or below the nodes (ML/BYPP). The tree is rooted to *Amniculicola immersa* CBS 123083 and *Amniculicola parva* CBS 123092. Newly generated sequences are highlighted in red, and the type strains are indicated in bold.

**Figure 2 jof-10-00319-f002:**
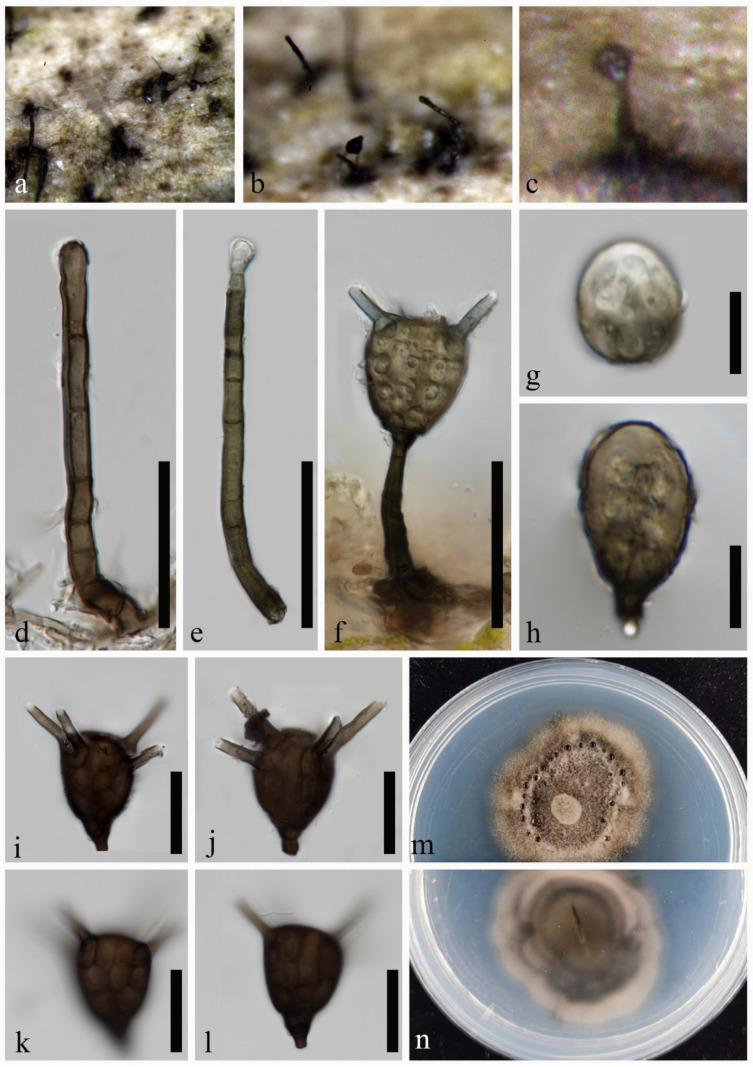
*Aquatisphaeria thailandica* (MHZU 24-0005). (**a**–**c**) Colonies on natural substratum. (**d**–**f**) Conidiophores, conidiogenous cells and conidia. (**g**–**l**) Conidia. (**m**,**n**) Colonies on PDA (front and reverse). Scale bars: (**d**–**f**) = 50 µm, (**g**–**l**) = 20 µm.

**Figure 3 jof-10-00319-f003:**
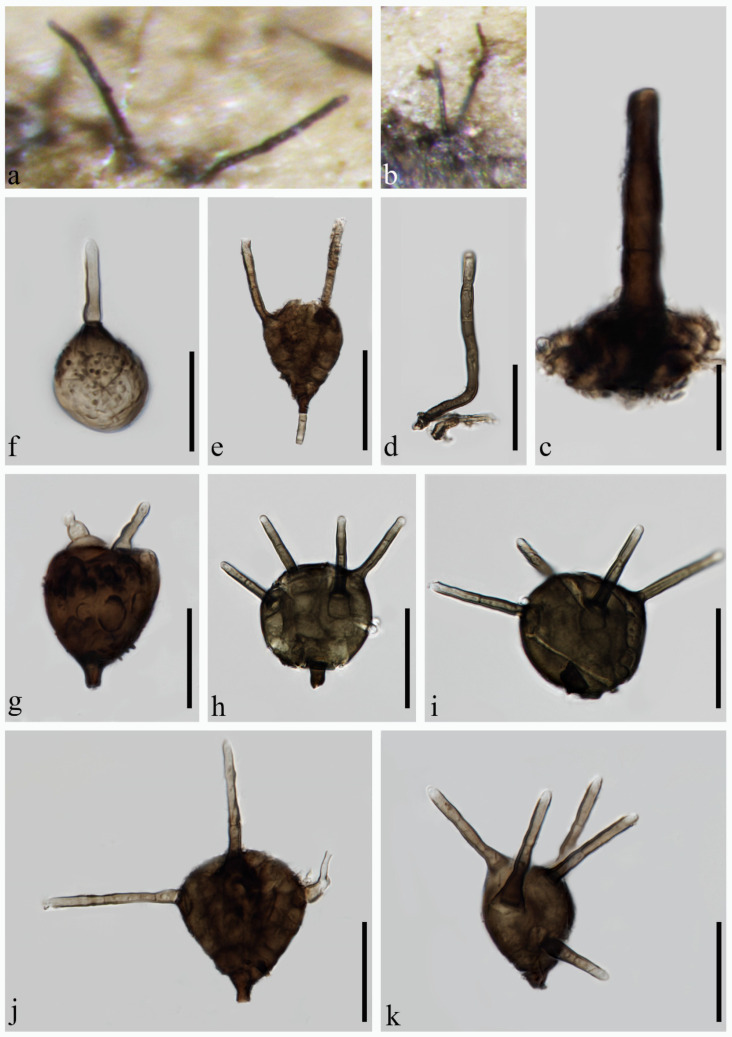
*Aquatisphaeria bambusae* (MHZU 24-0007, **holotype**). (**a**,**b**) Colonies on natural substratum. (**c**–**e**) Conidiophores, conidiogenous cells and conidia. (**f**–**k**) Conidia. Scale bars: (**c**,**d**) = 50 µm, (**e**–**k**) = 20 µm.

**Figure 4 jof-10-00319-f004:**
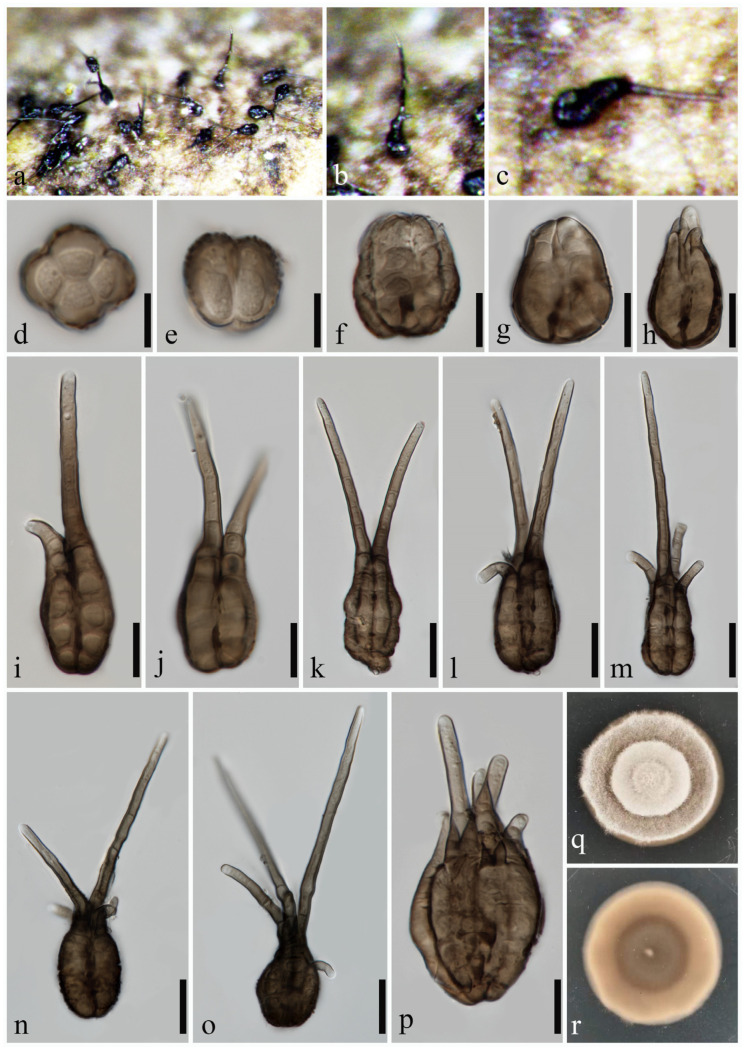
*Pseudotetraploa yangjiangensis* (MHZU 24-0008, **holotype**). (**a**–**c**) Colonies on natural substratum. (**d**–**p**) Conidia. (**q**,**r**) Colonies on PDA (front and reverse). Scale bars: (**d**–**p**) = 10 µm.

**Figure 5 jof-10-00319-f005:**
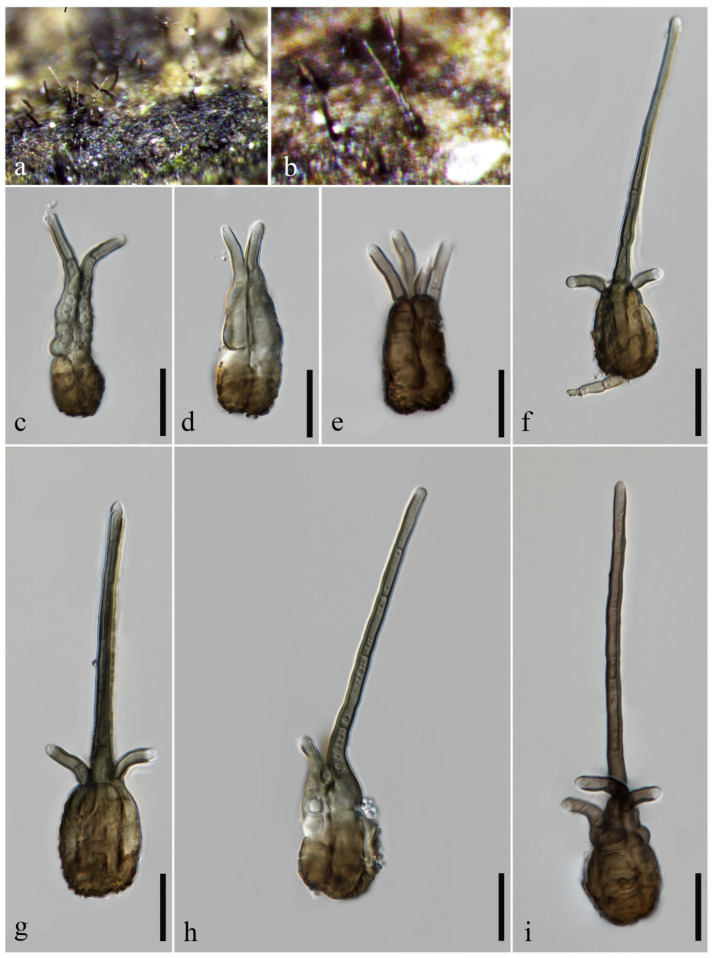
*Pseudotetraploa phyllostachydis* (MHZU 24-0006, **holotype**). (**a**,**b**) Colonies on natural substratum. (**c**–**i**) Conidia. Scale bars: (**c**–**i**) = 20 µm.

**Figure 6 jof-10-00319-f006:**
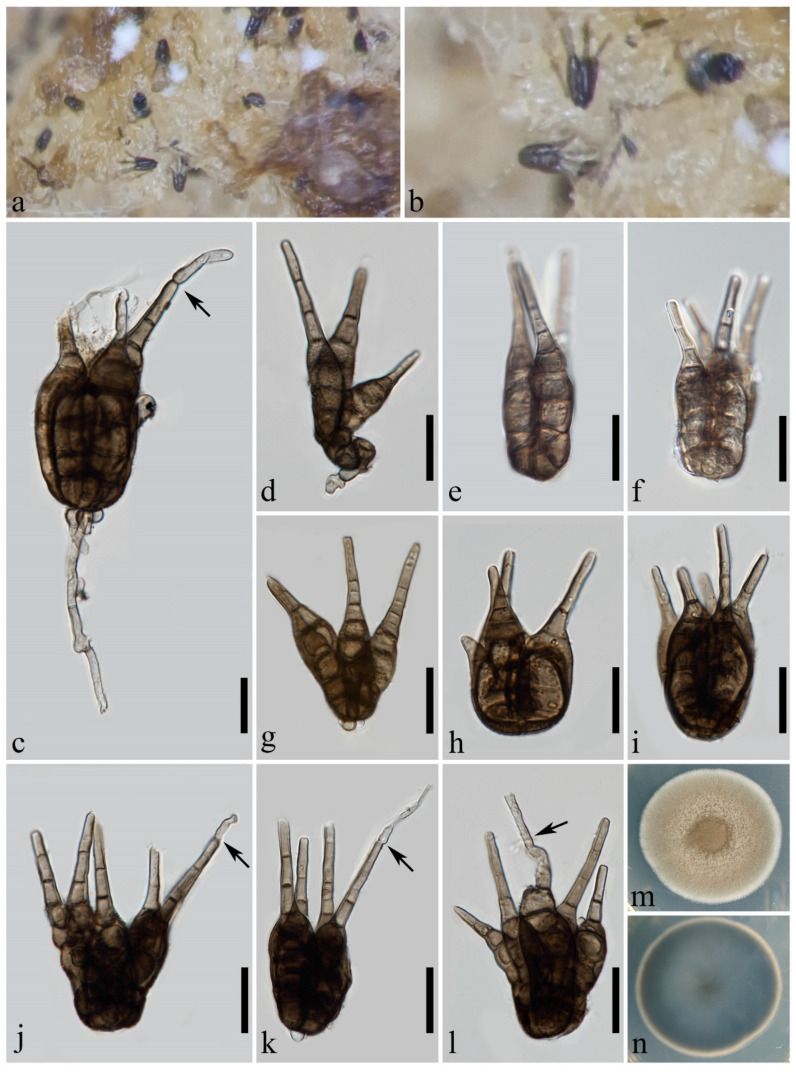
*Tetraploa submersa* (MHZU 24-0009, **holotype**). (**a**,**b**) Colonies on natural substratum. (**c**–**l**) Conidia (arrows indicate the ongoing growth of appendages). (**m**,**n**) Colonies on PDA (front and reverse). Scale bars: (**c**–**l**) = 20 µm.

**Table 1 jof-10-00319-t001:** Taxa used for phylogenetic analyses and their corresponding GenBank accession numbers. Newly generated sequences are indicated in red, ex-type strains are indicated in bold, and missing sequences are indicated with “N/A”.

Taxon	Strain	GenBank Accession Number				
		LSU	ITS	SSU	tub2	rpb2
** *Amniculicola immersa* **	**CBS 123083**	**NG_056964**	**N/A**	**NG_062796**	**N/A**	**N/A**
** *Amniculicola parva* **	**CBS 123092**	**NG_056970**	**N/A**	**NG_016504**	**N/A**	**N/A**
** * Aquatisphaeria bambusae * **	** ZHKUCC 24-0007 **	** PP336667 **	** PP336659 **	** PP336663 **	** PP346805 **	** N/A **
** *Aquatisphaeria thailandica* **	**MFLUCC 21-0025**	**MW890763**	**MW890969**	**MW890967**	**N/A**	**N/A**
*Aquatisphaeria thailandica*	DLUCC B151	MW890764	N/A	MW890968	N/A	N/A
* Aquatisphaeria thailandica *	ZHKUCC 24-0005	PP336665	PP336657	PP336662	PP346803	N/A
*Byssolophis sphaerioides*	IFRDCC 2053	GU301805	N/A	GU296140	N/A	GU456348
** *Ernakulamia cochinensis* **	**PRC 3992**	**LT964670**	**LT964671**	**N/A**	**LT964672**	**N/A**
*Ernakulamia cochinensis*	MFLUCC 18-1237	MN913716	MT627670	MT864326	N/A	N/A
** *Ernakulamia krabiensis* **	**MFLUCC 18-0237**	**MK347990**	**MK347773**	**MK347880**	**N/A**	**MK434872**
** *Ernakulamia tanakae* **	**NFCCI 4615**	**MN937211**	**MN937229**	**N/A**	**MN938312**	**N/A**
*Ernakulamia tanakae*	NFCCI 4616	MN937209	MN937227	N/A	MN938310	N/A
*Ernakulamia tanakae*	NFCCI 4617	MN937210	MN937228	N/A	MN938311	N/A
** *Ernakulamia xishuangbannaensis* **	**KUMCC 17-0187**	**MH260314**	**MH275080**	**MH260354**	**MN938311**	**N/A**
** *Ernakulamia cochinensis* **	**PRC 3992**	**MN913716**	**MT627670**	**MT864326**	**N/A**	**N/A**
*Ernakulamia cochinensis*	MFLUCC 18-1237	AB524604	AB524789	AB524463	AB524851	N/A
** *Polyplosphaeria fusca* **	**KT 1616**	**AB524604**	**AB524789**	**AB524463**	**AB524851**	**N/A**
*Polyplosphaeria fusca*	KT 1686	AB524606	N/A	AB524465	N/A	N/A
*Polyplosphaeria fusca*	KT 1640	AB524605	AB524790	AB524464	AB524852	N/A
*Polyplosphaeria fusca*	KT 1043	AB524603	AB5247988	AB524462	AB524850	N/A
*Polyplosphaeria fusca*	KT 2124	AB524607	AB5247991	AB524466	AB524853	N/A
** *Polyplosphaeria guizhouensis* **	**GZCC 23-0598**	**OR438888**	**OR427327**	**N/A**	**OR449118**	**N/A**
** *Polyplosphaeria hainanensis* **	**GZCC 23-0599**	**OR438889**	**OR427323**	**OR438285**	**OR449115**	**N/A**
*Polyplosphaeria hainanensis*	GZCC 23-0600	OR438890	OR427324	N/A	N/A	N/A
** *Polyplosphaeria nabanheensis* **	**KUMCC 16-0151**	**MH260312**	**MH275078**	**MH260352**	**MH412745**	**N/A**
** *Polyplosphaeria nigrospora* **	**ZHKUCC 22-0132**	**OR164963**	**OR164935**	**N/A**	**N/A**	**N/A**
** *Polyplosphaeria pandanicola* **	**KUMCC 17-0180**	**MH260313**	**MH275079**	**MH260353**	**N/A**	**N/A**
** *Polyplosphaeria thailandica* **	**MFLUCC 15** **–0840**	**KU248767**	**KU248766**	**N/A**	**N/A**	**N/A**
** *Pseudopolyplosphaeria guizhouensis* **	**GZCC 19-0247**	**OR209668**	**OR225074**	**OR134445**	**N/A**	**OR146944**
** *Pseudotetraploa bambusicola* **	**CGMCC 3.20939**	**ON332933**	**ON332915**	**ON332923**	**N/A**	**ON383991**
*Pseudotetraploa bambusicola*	UESTCC 22.0005	ON332934	ON332916	ON332924	N/A	ON383992
** *Pseudotetraploa curviappendiculata* **	**HHUF 28582**	**AB524608**	**AB524792**	**AB524467**	**AB524854**	**N/A**
*Pseudotetraploa curviappendiculata*	HHUF 28590	AB524609	AB524793	AB524468	AB524855	N/A
*Pseudotetraploa curviappendiculata*	KT 2558	AB524610	AB524794	AB524469	AB524856	N/A
*Pseudotetraploa javanica*	HHUF 28596	AB524611	AB524795	AB524470	AB524857	N/A
** *Pseudotetraploa longissima* **	**HHUF 28580**	**AB524612**	**AB524796**	**AB524471**	**AB524858**	**N/A**
** * Pseudotetraploa phyllostachydis * **	** ZHKUCC 24-0006 **	** PP336666 **	** PP336658 **	** N/A **	** PP346804 **	** PP346808 **
** *Pseudotetraploa rajmachiensis* **	**NFCCI 4618**	**MN937204**	**MN937222**	**N/A**	**MN938305**	**N/A**
*Pseudotetraploa rajmachiensis*	NFCCI 4619	MN937203	MN937221	N/A	MN938304	N/A
*Pseudotetraploa rajmachiensis*	NFCCI 4620	MN937205	MN937223	N/A	MN938306	N/A
** * Pseudotetraploa yangjiangensis * **	** ZHKUCC 24-0008 **	** PP336668 **	** PP336660 **	** N/A **	** PP346806 **	** PP346809 **
** *Pseudotetraploa yunnanensis* **	**KUNCC 10464**	**OR438891**	**OR449073**	**N/A**	**N/A**	**N/A**
** *Quadricrura bicornis* **	**CBS 125427**	**AB524613**	**AB524797**	**AB524472**	**AB524859**	**N/A**
** *Quadricrura meridionalis* **	**CBS 125684**	**AB524614**	**AB524798**	**AB524473**	**AB524860**	**N/A**
** *Quadricrura septentrionalis* **	**CBS 125430**	**AB524616**	**AB524800**	**AB524475**	**AB524862**	**N/A**
*Quadricrura septentrionalis*	CBS 125428	AB524617	AB524801	AB524476	AB524863	N/A
*Quadricrura septentrionalis*	CBS 125429	AB524615	AB524799	AB524474	AB524861	N/A
*Quadricrura septentrionalis*	CBS 125431	AB524618	AB524802	AB524477	AB524864	N/A
*Quadricrura septentrionalis*	CBS 125432	AB524619	AB524803	AB524478	AB524865	N/A
*Quadricrura septentrionalis*	CBS 125433	AB524620	AB524804	AB524479	AB524866	N/A
** *Shrungabeeja aquatica* **	**MFLUCC 18-0664**	**MT627663**	**MT627722**	**N/A**	**N/A**	**N/A**
** *Shrungabeeja fluviatilis* **	**GZCC 20-0505**	**OP377903**	**OP377804**	**OP377989**	**N/A**	**OP473080**
*Shrungabeeja fluviatilis*	GZCC 19-0511	MW133853	OP377851	MW134631	N/A	N/A
** *Shrungabeeja longiappendiculata* **	**BCC 76463**	**KT376472**	**KT376474**	**KT376471**	**N/A**	**N/A**
*Shrungabeeja longiappendiculata*	BCC 76464	KT376473	KT376475	N/A	N/A	N/A
*Shrungabeeja vadirajensis*	MFLUCC 17-2362	MN913685	MT627681	N/A	N/A	N/A
** *Tetraploa aquatica* **	**MFLU 19-0995**	**MT530452**	**MT530448**	**N/A**	**N/A**	**N/A**
*Tetraploa aristata*	CBS 996.70	AB524627	AB524805	AB524486	AB524867	N/A
** *Tetraploa bambusae* **	**KUMCC 21-0844**	**ON077067**	**ON077078**	**ON077073**	**ON075065**	**N/A**
** *Tetraploa cylindrica* **	**KUMCC 20-0205**	**MT893204**	**MT893205**	**MT893203**	**MT899417**	**N/A**
*Tetraploa cylindrica*	ZHKUCC 22-0087	ON555688	ON555689	ON555690	ON564477	N/A
** *Tetraploa dashaoensis* **	**KUMCC 21-0010**	**OL473555**	**OL473549**	**OL473556**	**OL505601**	**N/A**
** *Tetraploa dwibahubeeja* **	**NFCCI 4621**	**MN937208**	**MN937226**	**N/A**	**MN938309**	**N/A**
*Tetraploa dwibahubeeja*	NFCCI 4622	MN937206	MN937224	N/A	MN938307	N/A
*Tetraploa dwibahubeeja*	NFCCI 4623	MN937207	MN937225	N/A	MN938308	N/A
** *Tetraploa endophytica* **	**CBS 147114**	**MW659165**	**KT270279**	**N/A**	**N/A**	**N/A**
** *Tetraploa lignicola* **	**KUNCC 10794**	**ON422294**	**ON422286**	**ON422300**	**N/A**	**N/A**
*Tetraploa lignicola*	KUNCC 10795	ON422295	ON422287	ON422301	N/A	N/A
** *Tetraploa hainanensis* **	**GZCC 23-0601**	**OR438892**	**OR427325**	**OR438286**	**OR449116**	**N/A**
*Tetraploa hainanensis*	GZCC 23-0602	OR438893	OR427326	N/A	OR449117	N/A
*Tetraploa juncicola*	CBS 149046	ON603800	ON603780	N/A	N/A	N/A
** *Tetraploa nagasakiensis* **	**KT 1682**	**AB524630**	**AB524806**	**AB524489**	**AB524868**	**N/A**
*Tetraploa nagasakiensis*	KUMCC 18-0109	MK079891	MK079890	MK079888	N/A	N/A
** *Tetraploa obpyriformis* **	**KUMCC 21-0011**	**OL473554**	**OL473558**	**OL473557**	**OL505600**	**N/A**
** *Tetraploa pseudoaristata* **	**NFCCI 4624**	**MN937214**	**MN937232**	**N/A**	**MN938315**	**N/A**
*Tetraploa pseudoaristata*	NFCCI 4625	MN937212	MN937230	N/A	MN938313	N/A
*Tetraploa pseudoaristata*	NFCCI 4626	MN937213	MN937231	N/A	MN938314	N/A
** *Tetraploa puzheheiensis* **	**KUMCC 20-0151**	**MT627655**	**MT627744**	**N/A**	**N/A**	**N/A**
** *Tetraploa sasicola* **	**KT 563**	**AB524631**	**AB524807**	**AB524490**	**AB524869**	**N/A**
*Tetraploa sasicola*	FU 31019	MN937218	MN937236	N/A	N/A	N/A
*Tetraploa scheueri*	CY112	N/A	HQ607964	N/A	N/A	N/A
*Tetraploa* sp. 1	KT 1684	AB524628	N/A	AB524487	N/A	N/A
*Tetraploa* sp. 2	KT 2578	AB524629	N/A	AB524488	N/A	N/A
** * Tetraploa submersa * **	** ZHKUCC 24-0009 **	** PP336669 **	** PP336661 **	** PP336664 **	** PP346807 **	** N/A **
** *Tetraploa thailandica* **	**MFLUCC 21-0030**	**MZ412530**	**MZ412518**	**MZ413274**	**N/A**	**N/A**
** *Tetraploa thrayabahubeeja* **	**NFCCI 4627**	**MN937217**	**MN937235**	**N/A**	**MN938318**	**N/A**
*Tetraploa thrayabahubeeja*	NFCCI 4628	MN937215	MN937233	N/A	MN938316	N/A
*Tetraploa thrayabahubeeja*	NFCCI 4629	MN937216	MN937234	N/A	MN938317	N/A
** *Tetraploa wurfbainiae* **	**ZHKUCC 23-0954**	**OR626041**	**OR626039**	**N/A**	**OR653395**	**N/A**
*Tetraploa wurfbainiae*	ZHKUCC 23-0955	2R626042	OR626040	N/A	OR653396	N/A
** *Tetraploa yakushimensis* **	**KT 1906**	**AB524632**	**AB524808**	**AB524491**	**AB524870**	**N/A**
** *Tetraploa yunnanensis* **	**MFLUCC 19-0319**	**MN913735**	**MT627743**	**MT864341**	**N/A**	**MT878451**
*Tetraploa yunnanensis*	MFLUCC 18-0652	MN913697	MT627711	N/A	N/A	N/A
** *Triplosphaeria acuta* **	**KT 1170**	**AB524633**	**AB524809**	**AB524492**	**AB524871**	**N/A**
*Triplosphaeria cylindrica*	KT 2550	AB524636	AB524811	AB524495	AB524873	N/A
*Triplosphaeria cylindrica*	KT 1800	AB524635	AB524810	AB524494	AB524872	N/A
** *Triplosphaeria guizhouensis* **	**GZCC 19-0512**	**MW133854**	**OQ646060**	**MW134632**	**OQ659019**	**N/A**
** *Triplosphaeria maxima* **	**KT 870**	**AB524637**	**AB524812**	**AB524496**	**AB524874**	**N/A**
*Triplosphaeria* sp.	HHUF 27481	AB524815	AB524640	AB524499	AB524877	N/A
*Triplosphaeria* sp.	KT 2546	AB524641	AB524816	AB524500	AB524878	N/A
** *Triplosphaeria yezoensis* **	**KT 1715**	**AB524638**	**AB524813**	**AB524497**	**AB524875**	**N/A**
*Triplosphaeria yezoensis*	KT 1732	AB524639	AB524814	AB524498	AB524876	N/A

Abbreviations: BCC: Biotec Culture Collection, Bangkok, Thailand; BCRC: FU: Bioresource Collection and Research Center Collection, Taiwan, China; CBS: Westerdijk Fungal Biodiversity Institute, Utrecht, the Netherlands; CGMCC: China General Microbiological Culture Collection Centre, Beijing, China; DLUCC: Dali University Culture Collection, Yunnan, China; GZCC: Guizhou Culture Collection, Guizhou, China; HHUF: Herbaria of Hirosaki University; IFRDCC: Culture Collection, International Fungal Research and Development Centre, Chinese Academy of Forestry, Kunming, China; KUNCC: Kunming Institute of Botany Culture Collection; KT: Kazuaki Tanaka; MFLUCC: Mae Fah Luang University Culture Collection, Chiang Rai, Thailand; MFLU: Mae Fah Luang University Herbarium Collection; NFCCI: National Fungal Culture Collection of India NFCCI-A National Facility; PRC: the Herbarium of the Charles University, Prague, Czech Republic; UESTCC: University of Electronic Science and Technology Culture Collection, Chengdu, China; ZHKUCC: Zhongkai University of Agriculture and Engineering Culture Collection, Guangzhou, China.

## Data Availability

All data availability was mentioned in the manuscript. The novel taxa were registered in Index Fungorum (http://www.indexfungorum.org/Names/Names.asp, accessed on 24 February 2024), all taxa were submitted in Facesoffungi (http://www.facesoffungi.org, accessed on 24 February 2024) and the newly generated sequences were deposited into GenBank (https://www.ncbi.nlm.nih.gov/genbank/submit/, accessed 14 February 2024).

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
