# Peer review of "New Insights into Tetraplosphaeriaceae Based on Taxonomic Investigations of Bambusicolous Fungi and Freshwater Fungi"

_jof, 2024, doi:10.3390/jof10050319_

Round 1
Reviewer 1 Report
This is an interesting contribution in relation to fungi inhabiting bamboo and freshwater environments. It also addresses the status of Aquatisphaeria thailandica, a species that needs to be studied in detail because of its scarce occurrence in these environments.The work is excellently illustrate. Minor corrections are suggested in the attached pdf.
This is an interesting contribution in relation to fungi inhabiting bamboo and freshwater environments. It also addresses the status of Aquatisphaeria thailandica, a species that needs to be studied in detail because of its scarce occurrence in these environments.The work is excellently illustrate. Minor corrections are suggested in the attached pdf.

Reviewer 2 Report
The diversity of fungi remains poorly studied. Especially a lot of new data is hidden in tropical regions. This Study is devoted to the study of new representatives of a very interesting family of fungi, Tetraplosphaeriaceae, distributed mainly in bamboos and freshwater habitats. This group has been identified relatively recently and is being actively studied; data on their distribution, morphology and nucleotide sequences is accumulating. This information allows us to reconstruct the molecular phylogeny of this group.
In this work, a large amount of material was carefully analyzed. All requirements necessary for the description of new species have been met. Specimens were deposited in the herbarium, the living strain were deposited in the culture collection, the novel species were registered in the databases Index Fungorum and Faces of fungi.
Multi-locus phylogenetic analysis was carried out at a high methodological level. For each of the new species, 4–5 taxonomically significant regions of genomic DNA were analyzed. The phylogenetic relationships among 98 taxa of Tetraplosphaeriaceae have been extensively investigated using DNA sequence data.
As a result of this study, provides a new collections and new geographical record of one species and a description of 4 new species from the family Tetraplosphaeriaceae.
Detailed descriptions of these species on natural substrates and in pure culture are provided. Differences from morphologically similar species are analyzed. The species are beautifully illustrated with photographs of colonies on natural substrates, colonies on a culture medium, and microphotographs of the features of conidia.
Finally, there is an interesting discussion of the ecological preferences of species of the family Tetraplosphaeriaceae that have been isolated from bamboo substrates and freshwater habitats, accompanied by notes for each genus explaining the isolation source for those particular species.
The research was thoroughly planned, carried out at a high technical level, the material was analyzed and discussed, and the article was carefully prepared. It presents data on new discoveries and describes 4 species of fungi new to science from an interesting and relevant group for study. This work is certainly recommended for publication.
The authors done a very important observation about the connection between the two types of habitats through the taxonomic group of fungi. It would be interesting to supplement the data with information on the location of bambusicolous habitats where Tetraplosphaeriaceae species are recorded. Were they in riparian zone or dry? If such information is available.
